# Comment on Hermeling et al. Nano-Dry-Melting: A Novel Technology for Manufacturing of Pharmaceutical Amorphous Solid Dispersions. *Pharmaceutics* 2022, *14*, 2145

**DOI:** 10.3390/pharmaceutics17060714

**Published:** 2025-05-29

**Authors:** Dave A. Miller, Sandra U. Kucera, Daniel Ellenberger, Daniel Davis, Robert O. Williams

**Affiliations:** 1AustinPx, LLC, 111 W. Cooperative Way, Bldg. 3, Suite 300, Georgetown, TX 78626, USA; 2College of Pharmacy, The University of Texas at Austin, Austin, TX 78712, USA

## 1. Letter to the Editor

My colleagues and I are writing in response to an article published in the October 2022 issue of *Pharmaceutics* titled “Nano-Dry-Melting: A Novel Technology for Manufacturing of Pharmaceutical Amorphous Solid Dispersions” [1]. In Table 1 of this article, the authors, Hermeling et al. (2022) [1], list the advantages and disadvantages of several amorphous solid dispersion (ASD) manufacturing methods including the KinetiSol^®^ technology. Hermeling et al. state the disadvantages of KinetiSol as the following: having a “high mechanical load”, yielding a “less stable product”, and having a “high risk of drug degradation”. We strongly disagree with these stated disadvantages, as they are purely speculative and not supported by references. Moreover, the preponderance of the published record on KinetiSol directly contradicts these claims. The purpose of this comment is to rebut and correct the record by highlighting numerous published examples that refute the speculative statements of these authors about KinetiSol.

## 2. KinetiSol’s High Mechanical Load Is an Asset

We characterize the KinetiSol process in our peer-reviewed published papers as utilizing a high degree of mechanical energy. The unique internal design of KinetiSol equipment controls the distribution of this energy within the system. The assessment of this feature requires an accurate understanding of KinetiSol equipment’s design and mixing dynamics, which Hermeling et al. lack. Furthermore, numerous conventional pharmaceutical unit operations are associated with high mechanical load, e.g., various high-energy milling methods, high-shear granulation, roller compaction, and tablet compression, without detriment to the processed materials. Hermeling et al. did not consider these as precedents for the safe application of energy-intensive unit operations in pharmaceutical manufacturing when arbitrarily categorizing KinetiSol’s high mechanical load as a disadvantage.

High mechanical load is not in itself a disadvantage but rather only a liability if it causes the degradation of processed materials. Throughout KinetiSol’s extensive peer-reviewed publication and development history, this technology has been proven to be capable of processing a wide variety of drug compounds [2,3,4,5] and pharmaceutical excipients [6,7,8,9] without compromising their chemical purity, thereby demonstrating that high mechanical load, when judiciously administered via precise engineering, is not a disadvantage. It is instead a differentiating attribute enabling KinetiSol to produce ASD compositions unattainable by other technologies. The details and references supporting the above statements will be further elaborated in the paragraphs to follow.

## 3. KinetiSol Products Are Stable

The most inaccurate of the KinetiSol disadvantage statements made by Hermeling et al. is that KinetiSol produces a “less stable product”. First, the authors provide no citations supporting this statement, an immediate indication that this is purely speculation and completely unsubstantiated. Since the first KinetiSol peer-reviewed publication in 2009, this technology has been repeatedly shown to produce ASD compositions that are as molecularly uniform and stable, if not more stable, than those of alternative ASD technologies. Evidence regarding the exceptional stability of KinetiSol products is found not only in peer-reviewed research articles but also in the patent literature and regulatory filings, as detailed below.

Fundamental to the physical stability of an ASD system is the formation of a molecularly mixed, single-phase composition with a high glass transition temperature (T_g_). In 2009, DiNunzio et al. first reported the use of KinetiSol for the thermal production of a single-phase ASD of itraconazole (ITZ) in hypromellose (HPMC) [10]. This result was significant as the ITZ:HPMC ASD was documented by Six et al. in 2003 and confirmed by DiNunzio et al. in 2009 to be multi-phase and presumably less stable when processed by HME [11]. DiNunzio et al. in a 2010 article demonstrated an increased T_g_ (101 °C) and consequently superior physical stability of an ITZ–Eudragit L100-55 ASD produced by KinetiSol relative to the same composition made by HME that required a plasticizer to enable processing (T_g_ = 54 °C) [7]. The HME composition exhibited drug recrystallization after 6 months at 40 °C/75% RH, while the KinetiSol composition remained amorphous. Bennet et al. conducted a study in 2013 comparing the ASDs of acetyl-11-keto-beta-boswellic acid (AKBA) prepared by KinetiSol and solvent evaporation [12]. These authors showed that after 12 months of storage at ambient conditions, drug recrystallization occurred in the solvent-evaporated ASDs but not in the KinetiSol-processed ASDs. In a 2016 article, Brough and coworkers published results showing the 30-month physical stability of a KinetiSol-enabled ITZ–polyvinyl alcohol ASD [13]. In 2018, Jermain et al. applied a variety of analytical techniques, including solid-state NMR (ssNMR), to KinetiSol ASD formulations of varying drug loadings to demonstrate that these formulations were “homogenously and molecularly well dispersed” [14]. Similarly, in a 2020 article, Jermain et al. once again applied a variety of analytical techniques, including ssNMR, to compare the molecular arrangement of identical ASD formulations prepared by KinetiSol and spray drying. These authors confirmed that the KinetiSol ASD and SDD systems were both “molecularly miscible” [15]. We and our collaborators have also published numerous examples of the superior in situ amorphous stability of KinetiSol ASDs relative to that of spray-dried dispersions (SDDs) and micro-precipitated bulk powder (MBP), resulting in improved bioavailability [3,15,16]. This effect was concluded to be the result of a molecularly uniform ASD with a dense particle structure that delays water ingress and the destabilization of the amorphous system relative to high-surface-area solvent-generated particles.

As discussed in a review article by Ellenberger et al. (2018) [2], KinetiSol was applied to the compound deferasirox (DFX) to produce a high-dose (360 mg) immediate-release tablet containing a high-drug-load ASD (DST-0509). Despite DFX’s elevated melting point (265 °C), KinetiSol was successful at generating a single-phase ASD at 50% drug loading with purity equal to that of the bulk drug substance (99.5%). Having recently completed a phase II clinical trial [17], a substantial stability package exists for DST-0509 demonstrating excellent physical and chemical stability, with real-time stability extending out for several years. KinetiSol was also successfully applied to abiraterone (ABR) for the development of a more bioavailable, and consequently more efficacious, drug product (DST-2970) to improve patient outcomes in metastatic prostate cancer [4,18]. DST-2970 has been the subject of various regulatory filings, culminating in a recently concluded phase 1b study [19] and thus, like DST-0509, is supported by a robust stability package. These are just two examples of many clinical-stage drug products that have been developed by employing the KinetiSol platform and have been, by regulatory necessity, demonstrated to be pure and stable.

In summary, according to the results generated per the current state of the art in solid-state characterization techniques and in-depth product stability testing for numerous KinetiSol ASD-based drug products, it has been established that KinetiSol ASD systems are homogenously and molecularly mixed, thus resulting in highly stable drug products. Hermeling et al. either failed to adequately review the KinetiSol literature or simply ignored these results in their speculative characterization of KinetiSol as yielding a “less stable product”.

## 4. KinetiSol Substantially Reduces Risk of Drug Degradation

Interestingly, Hermeling et al. list “highly reduced processing times compared to HME” and “lower temperatures (than HME)” as advantages of KinetiSol, yet they state the “high risk of drug degradation” as a disadvantage of the technology but do not list this disadvantage for HME. This seems to imply that time and temperature are irrelevant to drug degradation for thermal-based ASD processing (clearly not true) and suggests that mechanical energy input is the primary concern for drug degradation. We already addressed the irrelevance of high mechanical load, so this will not be further discussed here. Noting that Hermeling et al. did not cite even one of our peer-reviewed papers, it is difficult to determine on what basis their claim of the “high risk of drug degradation” was made. Rather, Hermeling et al. cited a review article by Bhujbal et al. (2021) [20] from which we can only infer, based on the dismissal of time and temperature advantages, that the discussion of processing speed’s effect on the degradation of ritonavir (RTV) from LaFountaine et al. [9] was the single piece of information that served as the basis for this inference.

First, Bhujbal et al. (2021) [20] incorrectly reported that LaFountaine et al. evaluated RTV ASDs in copovidone (PVPVA) when in fact LaFountaine et al. actually only reported their results using the polymer polyvinyl alcohol (PVA). This is important because PVA (semi-crystalline) and PVPVA (amorphous) have completely different thermo-mechanical properties, with PVA presenting a much greater challenge for the thermal processing of the temperature- and shear-sensitive RTV. In fact, prior to the advent of KinetiSol, PVA had never been used as the primary carrier in an ASD formulation because the polymer’s semi-crystalline nature and high melt viscosity were prohibitive to HME, and this polymer is not soluble in organic solvents. The RTV-PVA ASD represented a novel and highly challenging composition with respect to drug–polymer compatibility for a thermal ASD process, yet LaFountaine et al. successfully generated acceptable ASDs of this binary system. The process development data presented by LaFountaine et al. showed the sensitivity of this specific composition (shear-sensitive drug and highly viscous polymer) to rotational speed in the context of process optimization for the identification of optimum parameters, which were identified by the study. This information was thus taken out of context by Hermeling et al. in that they selectively chose data from outside of the viable processing space for one specific composition to incorrectly generalize about the entire KinetiSol platform, ignoring the concluding comments of Bhujbal et al. (2021) [20] regarding this study as demonstrating “the capacity of KinetiSol for generating ASDs after adequate process optimization”. Furthermore, two subsequent studies illustrated the successful production of RTV ASDs by KinetiSol with a reduced impurity profile relative to the commercial Norvir^®^ tablet despite processing with HPMCAS-M, an acidic, less RTV-compatible polymer [21,22].

Furthermore, KinetiSol has been shown repeatedly in the peer-reviewed literature to produce viable ASD compositions with thermally labile compounds [2,4,5,8,9,22,23,24,25]. Often, these studies show a substantial reduction in or elimination of thermal degradation relative to HME. In other cases, HME was not sufficiently viable to even provide a comparison. KinetiSol, which boasts significant reductions in processing times and temperatures relative to HME, as correctly pointed out by Hermeling et al., is the most successful non-solvent ASD process for avoiding drug degradation. Considering the increasing frequency of aqueous and organic insoluble compounds in development and the recent movement toward “green chemistry” [26], KinetiSol is emerging as a principal ASD technology. Admittedly, KinetiSol does present a somewhat greater risk of thermal degradation than spray drying; however, we have only encountered rare cases where a drug is not KinetiSol-compatible, and solvent processing is the only option. The published record is thus clear about the utility of KinetiSol in producing ASDs with chemically labile compounds. For Hermeling to claim drug degradation that is a disadvantage of KinetiSol is to completely disregard a substantial body of peer-reviewed published evidence to the contrary.

## 5. Conclusions

In conclusion, we feel that the preponderance of the peer-reviewed published record supports the use of KinetiSol as a commercially relevant, non-solvent process for producing pure, single-phase, stable ASDs. The statements made by Hermeling et al. regarding the disadvantages of KinetiSol are unfounded based on our extensive history of publications, of which none were cited by Hermeling et al. These statements potentially mislead readers toward prematurely rejecting KinetiSol while bolstering interest in their technology. Furthermore, we would like to point out that there is no shortage of challenging, insoluble compounds in development, and our industry needs a variety of solutions to solve these problems. There is ample opportunity for all to succeed in this space and thus no need to misrepresent the innovations of others without fairly reviewing the published literature.

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
