# Peer review of "Comment on Hermeling et al. Nano-Dry-Melting: A Novel Technology for Manufacturing of Pharmaceutical Amorphous Solid Dispersions. *Pharmaceutics* 2022, *14*, 2145"

_pharmaceutics, 2025, doi:10.3390/pharmaceutics17060714_

Round 1
Reviewer 1 Report
Comments and Suggestions for Authors
Miller et al. has supported their argument with suitable justification and supporting literature. All the comments by Miller et al. and their group are acceptable, with no additional comments.
Author Response
Agreed. No response.
Reviewer 2 Report
Comments and Suggestions for Authors
The reviewer agrees with the authors'comment to the article published in Pharmaceutics. The explanation described by the authors is very clear and logical. Reference 25 was not mentioned in the text but was listed in Reference Section. "25" written in the Section might be deleted.
Author Response
Agreed. Deleted 25 in the References section.
Reviewer 3 Report
Comments and Suggestions for Authors
The authors, in this comment to the article by Hemerling et al. on the nano-dry-melting technology, attempt to respond to certain remarks that appear to be unfounded, providing relevant literature that supports their views. Considering the points of concern of the Hemerling paper, it is my opinion that there are indeed some hasty conclusions and unfounded statements in the Hemerling article. The points raised by the authors in their comment are valid, but I have an objection regarding the phrase "...and thus no need to misrepresent the innovations of others without fairly reviewing the published literature" in the conclusions, as it implies -or could be interpreted as such- that the misrepresentation of the KinetiSol technology was intentional. I would rather see this part of the text rephrased, to avoid any such implications or misinterpretations.
Author Response
Thank you for sharing your perspective on the phrasing of the final sentence in the Conclusion section. We appreciate your feedback.
Regarding the strong wording used, we acknowledge your concerns. However, we would like to emphasize that our intention was to highlight the potential reputational damage that could be caused to novel technologies as a result of the introduction of speculative statements into the literature. It is worth noting that the authors consciously decided to publish their speculation without providing any supporting evidence or citations.
Reviewer 4 Report
Comments and Suggestions for Authors
Authors should add in a note at the end of their comment article (though it is implied in the address details) that they are a commercial entity responsible for the development and applications of Kinetisol technology.
Author Response
The authors agree with the comment to add a disclaimer indicating our association with AustinPx. We look to the Jornal to advise how best to incorporate this statement into the final published form.